# (Low) Energy Availability and Its Association with Injury Occurrence in Competitive Dance: Cross-Sectional Analysis in Female Dancers

**DOI:** 10.3390/medicina58070853

**Published:** 2022-06-26

**Authors:** Dasa Prus, Dragan Mijatovic, Vedran Hadzic, Daria Ostojic, Sime Versic, Natasa Zenic, Tatjana Jezdimirovic, Patrik Drid, Petra Zaletel

**Affiliations:** 1Faculty of Sport, University of Ljubljana, 1000 Ljubljana, Slovenia; dasa.prus@fsp.uni-lj.si (D.P.); vedran.hadzic@fsp.uni-lj.si (V.H.); petra.zaletel@fsp.uni-lj.si (P.Z.); 2Faculty of Health Studies, University of Mostar, 68000 Mostar, Bosnia and Herzegovina; dragan.mijatovic@fzs.sum.ba; 3Faculty of Medicine, University of Mostar, 68000 Mostar, Bosnia and Herzegovina; dariaostojic@gmail.com; 4Faculty of Kinesiology, University of Split, 21000 Split, Croatia; simeversic@gmail.com; 5Faculty of Sport and Physical Education, University of Novi Sad, 21000 Novi Sad, Serbia; tatjanaj.ns@gmail.com (T.J.); patrikdrid@gmail.com (P.D.)

**Keywords:** nutritional availability, modern dance, soft tissue, bone, anthropometry

## Abstract

*Background and objectives:* The risk of low energy availability is related to various health problems in sports. This cross-sectional study aimed to identify a possible association between various dance factors, anthropometrics/body build, and energy availability with injury occurrence in contemporary dancers. *Materials and Methods:* The participants were 50 female competitive dancers (19.8 ± 4.1 years of age). The independent variables included age, dance factors (amount of training and competitions per week–exposure time, experience in dance), anthropometrics/body composition (body height, mass, BMI, body fat percentage (BF%), and fat-free mass (FFM)), and energy availability score (EAS; evaluated by accelerometer-based measurement of energy expenditure and Dance Energy Availability Questionnaires). The dependent variables were the occurrence of (i) soft-tissue injuries and (ii) bone injuries. The measurements were obtained by experienced technicians during the pre-competition period for each specific dance discipline. Univariate and multivariate logistic regressions were calculated to identify the associations between independent variables and injury prevalence. *Results:* The results showed that EAS (OR = 0.81, 95% CI:0.65–0.91), age (OR = 1.65, 95% CI: 1.1–2.46), higher BF% (OR = 1.23, 95% CI: 1.04–1.46) and BMI (OR = 1.61, 95% CI: 1.05–2.47) were correlated with soft-tissue injuries. Dancers who suffered from bone injuries reported higher exposure time (OR = 1.21, 95% CI: 1.05–1.37) and had lower values of FFM (OR = 0.73, 95% CI: 0.56–0.98). Multivariate regression analyses evidenced a higher likelihood of soft-tissue injuries in older dancers (OR = 1.75, 95% CI: 1.21–2.95) and the ones who had lower EAS (OR = 0.84, 95% CI: 0.71–0.95) while the exposure time was associated with a higher likelihood of bone injuries (OR = 1.21, 95% CI: 1.05–1.39). *Conclusions:* In order to decrease the injury prevalence among dancers, special attention should be paid to maintaining adequate nutrition that will provide optimal available energy for the demands of training and performing. Additionally, the control of training volume should be considered in order to reduce traumatic bone injuries.

## 1. Introduction

Dancing, in general, represents a form of activity with both physical and artistic demands. On the one hand, the aesthetic component is emphasized, but in order to perform it efficiently and in accordance with specific dance principles and sports rules, a high level of fitness is also required [1]. Therefore, dancers, in general, have high training loads, and most of them begin specific training at younger ages. Therefore, accumulated load and stress inevitably result in acute and chronic musculoskeletal problems, just as in other sports [2,3]. Indeed, studies confirmed that musculoskeletal injuries represent a major medical issue among dancers and have a high level of occurrence, with prevalence spanning from 40% to 84% [2,4]. In particular, body sites with the highest risk of injury are the lower extremities and back; as for the type of injuries, soft-tissue injuries and overuse problems occur most often [2]. 

Several studies have examined the injury risk factors in dance [2,5,6]. Among other factors, researchers have highlighted internal (i.e., age, sex, anthropometry characteristics, the level of fitness capacities) and external factors (i.e., the style of dance, dance setting, choreography demands) as predictors of potential musculoskeletal injuries [5,7,8,9,10]. For example, Evans, Evans, and Carvajal [5] carried out a survey on 260 West End dance performers and found that the most significant predictors for injury were female sex, a history of previous injuries, missed performances due to previous injuries, more physically demanding roles, and performing on raked (angled) stages [5]. A cross-sectional study on a large sample of dancers from multiple styles found that age and body weight were associated with injury in jazz/contemporary dance, body height in classical ballet and folk dance, and training volume in classical ballet and jazz/contemporary dance [11]. Biological sex specifics can also contribute to a greater risk of injury. In particular, one study examining menstrual dysfunctions in 98 dancing students showed that oligomenorrheic and amenorrhoeic students had a lower body mass and a higher incidence of musculoskeletal injuries and chronic orthopedic problems compared to eumenorrheic ones [8]. Finally, one longitudinal study analyzed 500 dance injuries reported by 644 dancers during a two-year period [6]. The results showed that changes in mood (body dissatisfaction, perfectionism) and diet habits (drive for thinness, bulimic tendencies) were significantly greater in injured than in non-injured dancers [6].

Nutritional habits and optimal energy intake are crucial in the majority of sports activities in order to maintain the level of performance, assure optimal recovery and avoid fatigue, and consequently decrease the risk of injury occurrence and illness [12,13,14,15,16]. However, what is specific to dance sports is that dance is a performing art and thus puts pressure on the athletes/dancers/performers to maintain optimal body composition, low body weight, and aesthetic figure. This environment represents a specific risk for consequential health problems [17]. In particular, dancers very often do not meet the nutritional requirements necessary to endure extensive dance training (5–13 h/week), which consequently results in energy deficiency and low energy availability [18,19]. 

Low energy availability (LEA) can be defined as an individual’s imbalance between energy intake and energy expenditure due to exercise [20,21]. An athlete may be aware of a lack of energy at that moment, but the long-term effects of this condition are much more serious. A chronic LEA state causes several physiological adaptations that can lead to serious health-related issues [22]. Menstrual cycle disorders, bone and muscle health problems, changes in basal metabolic rate, immune resistance, and mental health problems are all recognized as consequences of LEA [23,24]. Its severity is clearly seen in the fact that it is part of the female athlete triad, along with hypothalamic amenorrhea and osteoporosis, established by the American College of Sports Medicine [25]. Previously, LEA was examined as a factor of potential influence on injury occurrence in sports, and results showed that it has a large impact on health and performance [26,27]. For example, a study on elite distance runners showed that LEA has a large impact on bone injury rates [26]. However, there is an evident lack of knowledge on the association that may exist between LEA and injury occurrence in dance sports, despite the fact that LEA is frequent in dancers. 

From the previous literature overview, it is clear that musculoskeletal injuries are important health-related problems in dance [2]. Additionally, due to the specificity of dance as an aesthetic sport, studies confirmed a high prevalence of energy deficiency and LEA in dance disciplines [18,19]. Finally, studies in other sports found evidence that LEA is a risk factor for injury occurrence in sports [26]. On the other side, there is a clear lack of research examining the association between LEA and injury occurrence among professional top-level dancers from various disciplines. Therefore, the main aim of this study was to determine associations between musculoskeletal injuries and (low) energy availability among professional dancers. Additionally, we observed anthropometric/body composition and dance-related factors as factors potentially related to injury occurrence. Initially, we hypothesized that LEA will be associated with higher injury occurrence, irrespective of dance style.

## 2. Materials and Methods

### 2.1. Participants and Design of the Study

Participants in this cross-sectional study were 50 Slovenian female dancers from multiple contemporary dance disciplines (rock and roll, hip hop, breakdance, jazz dance). The sample was collected in cooperation with the Dance Association of Slovenia (DAS). On the basis of the number of registered modern dancers in DAS, a confidence of 95%, error of 0.05, and known injury occurrence of 10%, the required sample for this research was 71 (Statistica, Tibco Software Inc., Palo Alto, CA, USA). However, due to the fact that our study was performed during a period of COVID-19 pandemic (later 2021–early 2022) and included direct measurement of anthropometrics and physical activity, please see later for details), we were not able to collect the specified number of participants. All participants signed informed consent waivers before the start of the measurements and were registered with a self-chosen code to ensure the protection of data privacy. For underage participants, consent was signed by at least one parent after being informed by DAS. There were several inclusion criteria: participants had at least 8 years of dance training and had regularly reached the highest national competition level. Dancers who experienced illness or injury, who were prescribed medications, or who were pregnant were not included in the study. This study was designed in accordance with the Helsinki–Tokyo Declaration and was authorized by the National Medical Ethics Committee (no. 0120-221/2021/3).

### 2.2. Variables and Measurement

Variables in this study included age (in years), dance factors (experience in dance, exposure time–number of training/competition hours per week), anthropometric/body composition indices, energy availability, and injury occurrence. 

The measurements were obtained during the pre-competition period for each specific dance discipline. Groups of no more than ten dancers were invited to the Faculty of Sports (University of Ljubljana), where they were informed about all the details of the testing protocol. 

The anthropometric variables included body height, which was measured while the participant was standing upright against a portable anthropometer (GPM, Zurich, Switzerland) (barefoot, in 0.1 cm increments); body mass was measured with a digital Seca 769 scale (Seca GmBH & Co. Kg, Hamburg, Germany). The body composition (body fat percentage—BF%, and fat-free mass—FFM) was measured with the BioScan 920-II device (Maltron International Ltd., Rayleigh, U.K.). The Maltron Bioscan 920-II was previously validated for the measurements of body composition and muscle mass for both whole-body and particular body parts in different populations [28,29]. All measurements were performed by an experienced technician, one of the authors of the study. 

All the dancers received an Actical physical activity monitor (Koninklijke Philips NV, Amsterdam, the Netherlands) to estimate their daily energy expenditure and had to wear it for 7 days [30]. In addition, they received instructions for recording food intake with the mobile application “See How You Eat Diary App” (Health Revolution Oy Ltd., Kotka, Finland). The examiners introduced the app to the participants and explained to them how to record daily meals. Energy intake was entered into the “Open Platform for Clinical Nutrition” (Department of Computer Systems, Jožef Stefan Institute, Ljubljana, Slovenia), which, based on food intake, enables the calculation of calories and the content of individual micronutrients and macronutrients in the diet. The energy availability score (EAS) was calculated on the basis of the established formula; EAS = (Energy Intake–Energy Exercise Expenditure)/FFM [31]. On the basis of the obtained score, we sorted dancers into three groups: low EA (<30 kcal·(kg/FFM)·day^−1^), reduced EA (30–45 kcal (kg/FFM)·day^−1^), and sufficient EA (>45 kcal·(kg/FFM)·day^−1^) [32].

Data on injuries were obtained through the Dance Energy Availability Questionnaire (DEAQ). In the first phase, DEAQ was translated into Slovenian, and for the purpose of this study, we used it to obtain data on: (i) sociodemographics (age, sex), (ii) dance-related variables (experience in dance (in years), dance style, volume of training/competitions per week (in hours)), and (iii) epidemiology and the history of injuries [33]. Specifically, for the occurrence of soft-tissue injuries, the subjects were asked to mark the frequency of injury in each body location (i.e., toes, foot, ankle, tibia, knee) and the type of injury (i.e., sprain, strain, rupture of the muscle/ligament/tendon, contusion abrasion, bursitis/tendinitis) which occurred over the last year. Dancers reported bone injuries for the whole dance career by marking the number of fractures and/or stress fractures in each location of the body. For the purpose of later analyses, injuries were observed on the binomial scale (non-injured vs. injured), separately for soft-tissue and bone injuries. 

### 2.3. Statistics

All the variables were checked for the normality of distribution using the Kolmogorov–Smirnov test, and descriptive statistics included means and standard deviations (for parametric variables) and frequencies and percentages (for non-parametric variables). 

Differences between groups based on injury occurrence were separately established by analysis of variance (ANOVA) for soft-tissue injuries and bone injuries. 

The association between independent variables (age, experience in dance, the volume of training and competitions/time of exposure, anthropometrics/body composition, energy availability) and injury occurrence were established by regression analyses. First, all independent variables were univariately correlated with binomial criteria (injured vs. non-injured) by logistic regression. In the second phase, all the independent variables found to be significantly associated with the injury occurrence were simultaneously included in the multivariate regression, which allowed us to control the associations for possible covariates. The odds ratio (OR) and 95% confidence interval (95% CI) were reported, while the Hosmer–Lemeshow test (HL) was used to evaluate the appropriateness of the model’s fit. 

Statistica version 13.5 (Tibco Inc., Palo Alto, CA, USA) was used for all analyses, and a *p*-level of 0.05 was applied.

## 3. Results

A total of 66% of the studied dancers reported the occurrence of soft-tissue injuries, with bursitis/tendinitis being the most prevalent (68% of dancers), followed by sprain (48%), abrasion (28%), rupture (of muscle, ligament or tendon) (20%), contusion (8%) and strain (reported by 6% of dancers).

With regard to the location of the soft-tissue injury, the most commonly injured locations were the ankle and knee (32% of dancers), followed by injured thighs (26%), toes (20%), and lumbar region (12%). 

Bone injury was reported by 16% of the studied dancers. Fracture was reported by 10% of dancers, while 6% of them reported stress fractures. The upper extremities were more commonly injured than the lower extremities when it came to bone injury (61% vs. 39%, respectively, among those dancers who reported bone injury). 

The 14 dancers (28%) were categorized in the low EA group (<30 kcal·(kg/FFM)·day^−1^), and 34 dancers (68%) were categorized in the reduced EA group (30–45 kcal·(kg/FFM)·day^−1^), while only 2 dancers (4%) had sufficient EA (>45 kcal (kg/FFM)·day^−1^). 

The descriptive statistics for all parametric variables are presented in Table 1. 

Dancers who reported soft-tissue injury were older (F = 10.11, *p* = 0.001), had longer dancing experience (F = 10.57, *p* = 0.001), and had higher BF%, BMI and lower EAS than non-injured dancers (F = 7.15, 5.57, *p* = 0.01 and F = 4.47, *p* = 0.03, respectively) (Table 2).

Those dancers who reported bone injury had higher exposure time per week (F = 14.25, *p* = 0.001) and had lower FFM (F = 4.49, *p* = 0.05) than non-injured dancers (Table 3). 

Univariate and multivariate correlates of soft-tissue injury occurrence are presented in Figure 1.

The univariate logistic regression calculated between the study variables and the occurrence of soft-tissue injury evidenced a higher risk for injury in older dancers (OR = 1.46, 95% CI: 1.10–2.46), those with higher BF% (OR = 1.23, 1.04–1.46), higher BMI (OR = 1.61, 95% CI: 1.05–2.47), and a lower EAS (OR = 0.81, 95% CI: 0.65–0.91) (Figure 1A). The multivariate regression model, which included all significant univariate predictors in a predictor set of variables, highlighted the dancers’ age (OR = 1.75, 95% CI: 1.21–2.95) and EAS (OR = 0.84, 95% CI: 0.71–0.95) as significant predictors of soft-tissue injury (Figure 1B), with appropriate model fit (HL = 0.57, *p* = 0.87).

Correlates of bone injury occurrence are presented in Figure 2.

The volume of training/competition per week (exposure time) was higher in dancers who reported bone injury (OR = 1.21, 95% CI: 1.05–1.37). Meanwhile, FFM was inversely related bone injury (OR = 0.73, 95% CI: 0.56–0.98) (Figure 2A). The multivariate model evidenced that exposure time was the only significant predictor of bony injury, with a higher risk of injury in dancers who spend more time training/competitions per week (OR = 1.21, 95% CI: 1.05–1.39; HL = 6.77, *p* = 0.56) (Figure 2B). 

## 4. Discussion

This study aimed to detect factors associated with musculoskeletal injuries in female contemporary dancers, paying special attention to the possible association between energy availability and injury occurrence. The results indicated several important things. First, soft-tissue injuries were more prevalent in older dancers and those with higher BF%, BMI, and lower EAS. Second, time of exposure was associated with higher injury occurrence, while higher FFM was correlated with a lower prevalence of bone injuries. Finally, when associations were evaluated throughout multivariate models, participants’ age was positively and EAS was negatively associated with the occurrence of soft-tissue injuries, while exposure time was negatively associated with the occurrence of bone injuries. 

### 4.1. Dancers’ Age and Injury Occurrence

Results of this study showed that the prevalence of soft-tissue injuries is higher in older dancers. Considering that dancers usually start practicing at early ages, we can assume that accumulated stress and load actually contribute to specific locomotor problems and consequent soft-tissue injuries. Highly demanding and strenuous dancing activity repetitively puts a load on the lower extremities in particular and increases the risk of acute and overuse injuries in this body region. This is confirmed in several previous studies that explored the prevalence of injuries and risk factors among dancers [7,34,35,36]. For example, a study on professional ballet dancers showed that the total number of injuries was directly associated with the dancer’s age [7]. Similar findings were observed in other sports, for both men and women, where authors regularly reported an increased rate of soft-tissue injuries along with the athlete’s age [37,38]. Therefore, our results of higher risk for injury occurrence in older dancers are supportive of some but not all previous studies.

Specifically, some studies in dance showed that the particular type of injuries decreases over the years [39]. For example, a study conducted on Swedish professional ballet dancers found that ankle sprain injuries were more frequent among younger dancers. Authors explained such findings as being due to improved technique, strength, and endurance over the years of specific dance training [39]. At the same time, such an explanation is not probable in our study simply because of the differences between ballet dance and the contemporary dance observed here. Namely, classic ballet techniques are learned from the very beginning of the career, and at the moment of joining the professional ensembles (approximately at the age of 18–19 years), all techniques should be mastered at an appropriate level. However, the improvement in technique occurs even later in the career, which consequently decreases the stress in the musculoskeletal system [39,40]. At the same time, dancers involved in modern dance are constantly exposed to more complex dance techniques and perform more demanding routines and more complex choreographies, which consequently increases the risk of injury occurrence irrespective of the improvement in their dancing technique over time [41,42]. 

### 4.2. Training Volume and Bone Injuries

Higher training volume (exposure time) was associated with the occurrence of bone injuries, as a higher number of training hours per day was significantly correlated with injury occurrence. Regarding traumatic injuries, this finding is simply explainable by an increased possibility of suffering traumatic injuries (i.e., fractures) among dancers who spend more time in training and performing. This is additionally aggravated by the negative influence of fatigue (as a result of time spent in training and performing). At the same time, overuse bone injuries, i.e., stress fractures, are probably related to higher training volume and load imposed on a dancer’s skeleton. Supportive findings were found in studies conducted three decades ago but also in newer ones [43,44]. 

For example, Kadel, Teitz, and Kronmal [44] investigated injuries among two ballet companies and evidenced that dancers who danced for more than 5 h per day were significantly more likely to have a stress fracture than those practicing less [44]. Moreover, a study on a large sample of dancers from different age categories (10 to 18 years of age) indicated that hours of training per week were significantly associated with injury occurrence [43]. Additionally, in support of previous consideration for the potential contribution of fatigue, the number of training hours was associated with fatigue-related injuries in dancers [6]. In brief, a higher training volume with an insufficient period of rest will lead to fatigue and consequent overtraining syndrome. Naturally, it altogether alters motor control, resulting in compensational and incorrect movement patterns, which can lead to both acute and traumatic bone injuries [45,46].

### 4.3. Anthropometric/Body Composition and Injury Occurrence

Higher FFM was negatively correlated to the occurrence of bone injuries. Since FFM includes both muscle and bone mass, this finding can be explained in two ways. Firstly, more muscle mass provides a higher level of muscle strength and better balance, which both allows the dancer to perform for a prolonged time while resisting fatigue. On the other hand, dancers with less muscle mass are less capable of coping with the exercise-related stress, and locomotor problems will occur more often. Although we could not find studies where anthropometric status was correlated with injury occurrence in dancers involved in investigated dance disciplines, there is some evidence of such associations in ballet dancers. 

Namely, research on full-time ballet students showed a negative correlation between mesomorphy (as a measure of muscle mass) and the number of overuse injuries sustained [47]. More specifically, dancers with more muscle mass, the dominantly mesomorph somatotype, recorded fewer injuries and had lower recovery time after they occurred [47]. Additionally, indirect support of the association between anthropometric status and injury occurrence in dance is found in another ballet study. In brief, Koutedakis et al. found that low thigh power outputs (which are directly related to muscle mass) are associated with the severity of low extremity injuries in ballet and contemporary dancers [48]. 

Although we have no direct insight into the exact proportions of the individual components, we can assume that a lower likelihood of suffering a bone injury was also related to higher bone mass (as one of the FFM components). Dancing is an activity where high loads are placed on the bones [40]. Meanwhile, a higher relative bone mass is positively correlated to bone density, which is known as an important protective factor against injuries in sports activities in general [49,50]. Therefore, we can assume that the association between FFM and bone injuries can be at least partially attributed to the previously explained mechanisms. 

The results showed a significant association between BF% and BMI and soft-tissue injuries. In particular, dancers with higher BF% and BMI values were more likely to become injured. These results are generally contrary to previous findings, as studies reported lower values of BF% and BMI as risk factors for musculoskeletal injuries [51,52]. For example, a study on female ballet dancers showed that ballerinas with a BMI lower than 19 are more prone to injuries than their peers with higher values [51]. Twitchett, Brodrick, Nevill, Koutedakis, Angioi, and Wyon [52] investigated the causes of injuries in dancers and found a significant negative correlation between BF% and injury occurrence [52]. The authors explained this to be a consequence of restrictive caloric intake, which results in reduced BF and BMI values, but also more proneness to injuries of the musculoskeletal system. However, in explaining the contradictory findings between our and previous studies (i.e., we have found body fat as factor of increased risk for injury occurrence, while previous studies reported opposite associations) once again we have to acknowledge the differences in the samples of participants between our and previous studies. Namely, in this study, we observed dancers involved in danc e disciplines, where BF% values are not so low as in previous studies conducted on ballet dancers (>19% and <16%, respectively). As a result, in our study, higher BF% and BMI values actually point to “higher ballast mass” which is known to be a risk factor for injury occurrence in sports settings [53,54]. 

### 4.4. (Low) Energy Availability and Soft-Tissue Injuries

Although the prevalence of LEA is a known problem in dance, and studies confirmed the association between LEA and injury occurrence in athletes, we did not find any study that examined the association between LEA and injury occurrence in dance [55]. However, our results clearly show the association between LEA and the occurrence of soft-tissue injury. In explaining these findings, a short overview of the possible co-associations is needed. 

In general, LEA is generally known to be associated with eating disorders [13,33,56]. At the same time, the association between eating disorders and musculoskeletal injuries was previously confirmed in multiple studies on different groups of female athletes [55,57,58]. Therefore, we can anticipate the association between LEA and injury occurrence as well. Additionally, the association between LEA and injury would be particularly possible in aesthetic sports such as dancing. Namely, female athletes involved in these sports are known to be under pressure to achieve and maintain an ideal body type, which is then associated even with previously mentioned eating disorders [33]. Putting it all together (high prevalence of low energy availability in our sample, possibility of eating disorder, association between eating disorder and injury occurrence), our results on the higher likelihood of being injured in dancers who have lower EAS scores are logical. 

At the same time, it is important to note that lower EAS was evidenced as being significantly associated with soft-tissue injuries, but we have found no association between EAS and bone injuries. The probable explanation is provided in the following text. First, it is relatively well documented that skeletal complications (including skeletal injuries) are related to eating disorders, but it is almost exclusively the case in starvations associated with various psychiatric illnesses (i.e., anorexia nervosa) and/or metabolic problems (i.e., amenorrhea) [59]. However, we studied young female athletes, meaning that such problems are not likely to occur. Second, bone injuries reported in our study were mostly acute and occurred after some traumatic events. Therefore, it is not likely that nutritional intake and available energy could be significantly associated with the occurrence of traumatic injury. On the other hand, negative energetic balance (the discrepancy between energy intake and caloric expenditure) actually highlights the fact that a dancer has an insufficient level of available energy needed to perform [13]. In these circumstances, the ability of the muscles to handle the required demands will be reduced while the risk of acute or chronic muscle-related problems will be higher.

### 4.5. Multivariate Associations

When all significant predictors were included in the multivariate logistic regression models, the contribution of the BF% and BMI did not reach statistical significance, while dancers’ age and EAS remained significant predictors of soft-tissue injuries. This can be explained simply by the fact that the BF% and BMI increase with age [60,61]. Knowing the previously discussed aesthetic component of dance, such physiological changes can naturally induce dancers toward low caloric intake (in order to control the BF% and BMI), which consequently results in LEA and injuries. In support of this, a cross-sectional study on the sample of one thousand female athletes ranging from 15 to 30 years old analyzed the differences between groups of athletes with adequate and low energy availability (AEA and LEA, respectively) [27]. The results showed that the LEA group was significantly older and had higher values of body weight and BMI compared to the AEA group [27].

Similarly, the multivariate model statistically diminished the protective effect of FFM on bone injuries as it was probably correlated to the hours of dance training. In other words, the total exposure time exceeds the positive impact of FFM and can be considered the most significant factor associated with the occurrence of bone injuries. Theoretically, it is also possible that dancers who spend more time in training actually try to decrease their BF values and increase FFM, which altogether may even explain the confounding effect of the FFM and the hours of training with regard to the occurrence of bone injuries in the studied participants.

### 4.6. Strengths and Limitations

One of the main limitations is the cross-sectional design of the study. Therefore, detecting causal relationships between the observed independent variables and injury occurrence may be limited. Additionally, injury data were collected retrospectively, and recall bias was possible. However, considering that we used previously validated and frequently used DEAQ as a measurement tool, the authors believe that the collected information can be considered appropriate. Finally, anthropometrics/body composition was measured once, while injury status was reported for the previous period. Therefore, there is a certain possibility that injury occurred at a moment when dancers’ had different anthropometric/body composition characteristics than those evidenced at measurement, and this was particularly possible for bone injuries that were reported for the whole dance career. 

We believe that the analysis of associations between the observed independent variables and injury prevalence will provide a certain contribution to the health safety among dancers, especially if we take into account that the sample of participants consisted of top national level dancers from different dance disciplines. This is one of the first studies that analyzed the problem of energy availability in several aspects of the female dance population. Knowing that LEA is common in aesthetic sports, which is confirmed even here, we hope that the study will initiate further research in the field.

## 5. Conclusions

The results evidenced a higher prevalence of bone injuries in dancers with longer exposure time, which is probably related to: (i) increased possibility of traumatic injury occurrence in dancers who spend more time in training and competition, (ii) negative influence of fatigue and the consequent lack of motor control in dancers who spend more time in training, and/or are more exposed to competitive efforts. 

At the same time, soft-tissue injuries are more prevalent in older dancers and the ones with lower EAS. Specifically, the imbalance between energy intake (caused by aesthetic criteria) and expenditure (high dancing demands) will cause a reduced ability of the muscular system to cope with the mechanical stress to which it is exposed and result in injuries. This energy imbalance will also have negative consequences when energy intake exceeds expenditure as an increased level of BF% and BMI will represent a significant risk for injuries, especially in older dancers. Therefore, our results highlight the importance of optimal nutrition and education among dancers on this issue as a crucial part of maintaining health and physical capacities.

## Figures and Tables

**Figure 1 medicina-58-00853-f001:**
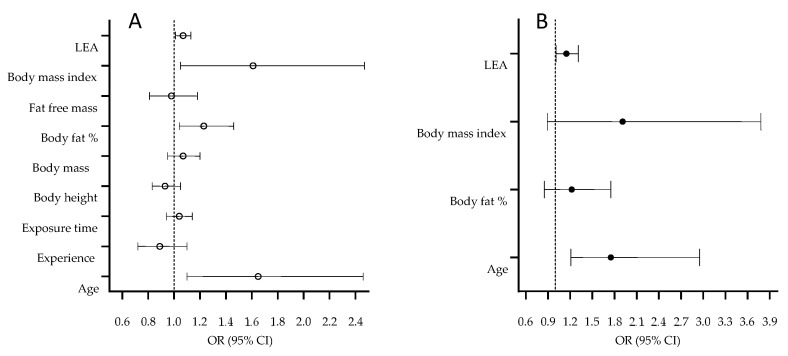
Univariate (**A**), and multivariate (**B**) correlates of soft-tissue injury in contemporary dance.

**Figure 2 medicina-58-00853-f002:**
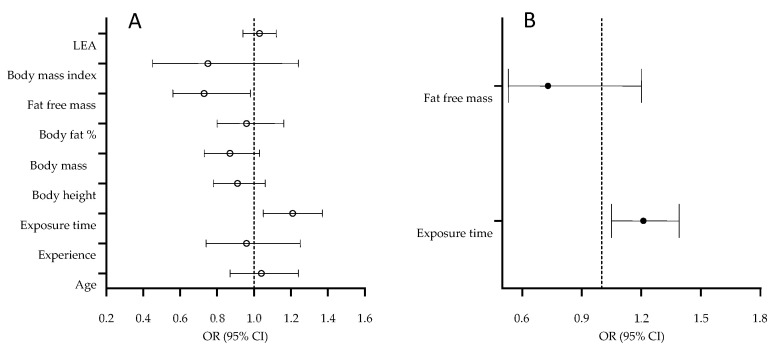
Univariate (**A**) and multivariate (**B**) correlates of bone injury in contemporary dance.

**Table 1 medicina-58-00853-t001:** Descriptive statistics for independent variables.

	Mean	Minimum	Maximum	Std. Dev.
Age (years)	19.86	15.00	33.00	4.05
Experience in dance (years)	11.32	8.00	17.00	2.45
Exposure time per week (hours)	15.85	8.00	39.50	6.76
Body height (cm)	165.95	154.40	179.10	5.09
Body mass (kg)	57.02	48.30	70.90	5.52
Body mass index (kg/m^2^)	20.68	17.50	25.70	1.70
Body fat (%)	19.31	10.89	29.86	4.20
Free fat mass (kg)	45.85	39.36	52.70	3.18
Energy availability score (kcal·(kg/free fat mass·day^−1^)	31.39	15.35	54.59	9.77

**Table 2 medicina-58-00853-t002:** Differences between groups based on soft-tissue injury occurrence.

	Injured (*N* = 33)		Non-Injured (*N* = 17)		ANOVA	
	Mean	Std. Dev.	Mean	Std. Dev.	F test	*p*
Age (years)	21.06	4.38	17.53	1.77	10.11	0.001
Experience in dance (years)	11.65	3.33	10.73	2.61	10.57	0.001
Exposure time per week (hours)	16.38	7.69	14.82	4.46	0.59	0.45
Body height (cm)	165.37	4.83	167.09	5.52	1.29	0.26
Body mass (kg)	57.67	5.46	55.75	5.57	1.36	0.25
Body mass index (kg/m^2^)	21.07	1.72	19.93	1.41	5.58	0.02
Body fat (%)	20.39	3.98	17.22	3.92	7.15	0.01
Free fat mass (kg)	45.78	3.33	45.98	2.97	0.04	0.83
Energy availability score (kcal·(kg/free fat mass·day^−1^)	27.89	12.62	33.20	7.54	4.47	0.03

**Table 3 medicina-58-00853-t003:** Differences between groups based on bone injury occurrence.

	Injured (*N* = 8)		Non-Injured (*N* = 42)		ANOVA	
	Mean	Std. Dev.	Mean	Std. Dev.	F test	*p*
Age (years)	20.38	5.04	19.76	3.90	0.15	0.70
Experience in dance (years)	10.63	6.12	9.67	4.85	0.24	0.63
Exposure time per week (hours)	23.19	9.72	14.45	5.10	14.25	0.001
Body height (cm)	163.93	4.19	166.34	5.19	1.53	0.22
Body mass (kg)	54.05	4.84	57.58	5.51	2.86	0.10
Body mass index (kg/m^2^)	20.06	1.78	20.80	1.67	1.29	0.26
Body fat (%)	18.80	4.56	19.41	4.18	0.14	0.71
Free fat mass (kg)	43.74	2.48	46.25	3.16	4.49	0.04
Energy availability score (kcal·(kg/free fat mass·day^−1^)	33.19	10.40	31.05	9.74	0.32	0.58

## Data Availability

The data are available upon reasonable request.

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
