# Peer review of "(Low) Energy Availability and Its Association with Injury Occurrence in Competitive Dance: Cross-Sectional Analysis in Female Dancers"

_medicina, 2022, doi:10.3390/medicina58070853_

Round 1

Reviewer 1 Report

Overall the manuscript provides insight into a noteworthy topic.

A few comments that should be attended to include:

Abstract:

- In line 30, the abstract reads like only older dancers with low EAS were at risk of soft tissue injury. But the results suggest that these were standalone factors as risk for injury. I think it needs clarifying to maintain consistency throughout the manuscript. A similar point is made in the conclusions in line 385, so again, consistency is needed.

- Line 81 in Intro, change 'the' before the word 'moment' to "that". Should read as "a lack of energy at that moment..."

Materials and Methods:

- Can the authors comment on the accuracy of the BioScan 920-II that was used to determine BF% and FFM?

- Authors need to make comment on the time period for which injuries were recalled/reported by study p'pants in the questionnaire.

- Were anthro variables only collected once during the study? If so, depending on the period of recall for the injuries reported, authors need to comment on how current body composition related to a retrospective injury. How can the authors be sure that the dancer had a similar physique at the time of injury as what they did at the time of taking anthro measures in the current study? This is perhaps a significant limitation also affecting the correlation of this factor to injury.

Results:

- In line 177, suggest amending to read "(68% of dancers)" and removing 'reported this type of injury'.

- Define 'rupture' - ie. of ligament or tendon? Line 178

- In line 181, suggest amending to read "(32% of dancers)" and remove 'reported injury on these two body parts'.

- In line 194, the results have been interpreted as dancers with longer dancing experience reported more soft tissue injury, however Table 2 shows that the injured dancers had mean experience of 10.73 years and non-injured had 11.65 years. This suggests that those with less experience were injured. Needs correction or clarification.

Discussion:

- In line 252, avoid the use of 'first person'. Replace "we" with third person terminology.

- The final comments made in section 4.1, lines 252-260 require a reference.

- The final comment made in section 4.2, lines 279-281 requires a reference.

- Line 320, remove additional full stop before "Low" (.Low)

- In line 337, suggest removing "(please see results...)"

- In section 4.6, the authors should avoid inferring that causation can be determined from their cross-sectional study. Whilst the discussion of these factors is plausible, this limitation should be reworded to ensure that it is known that a causal relationship has not been established from the cross-sectional nature of the study.

- An additional limitation regarding the link between injury data (depending on the period of recall) and current anthro measures needs to be acknowledged, even despite the fact that a valid instrument was used to collect injury data. 

Author Response

Overall the manuscript provides insight into a noteworthy topic.

A few comments that should be attended to include:

Thank You for all your suggestions. We followed them thoroughly and amended the manuscript accordingly.

Abstract:

- In line 30, the abstract reads like only older dancers with low EAS were at risk of soft tissue injury. But the results suggest that these were standalone factors as risk for injury. I think it needs clarifying to maintain consistency throughout the manuscript. A similar point is made in the conclusions in line 385, so again, consistency is needed.

Response: Thank You for noticing. The text is amended accordingly. In Abstract now it reads: “Multivariate regression evidenced higher likelihood of soft-tissue injuries in older dancers (OR = 1.75, 95%CI: 1.21-2.95) and the ones who had lower EAS (OR = 0.84, 95%CI: 0.71-0.95) while the exposure time was associated with a higher likelihood of bone injuries (OR = 1.21, 95%CI: 1.05-1.39).”

The conclusion section is systematically rewritten based on later suggestions and we paid attention on your concern.

- Line 81 in Intro, change 'the' before the word 'moment' to "that". Should read as "a lack of energy at that moment..."

Response: Thank You, the word ‘the’ is substituted with ‘that’.

Materials and Methods:

- Can the authors comment on the accuracy of the BioScan 920-II that was used to determine BF% and FFM?

Response: Thank You for your suggestion. The reference about the device validity is added: “The Maltron Bioscan 920-II was previously validated for the measurements of body com-position and muscle mass for both whole-body and particular body parts in different populations [28,29]”

  1. Sipers, W.M.; Dorge, J.; Schols, J.M.; Verdijk, L.B.; Van Loon, L.J. Multifrequency bioelectrical impedance analysis may represent a reproducible and practical tool to assess skeletal muscle mass in euvolemic acutely ill hospitalized geriatric patients. European Geriatric Medicine 2020, 11, 155-162.
  2. Longland, T.M.; Oikawa, S.Y.; Mitchell, C.J.; Devries, M.C.; Phillips, S.M. Higher compared with lower dietary protein during an energy deficit combined with intense exercise promotes greater lean mass gain and fat mass loss: a randomized trial. Am J Clin Nutr 2016, 103, 738-746, doi:10.3945/ajcn.115.119339.

- Authors need to make comment on the time period for which injuries were recalled/reported by study p'pants in the questionnaire.

Response: Thank you for noticing it, indeed, we missed to report it originally. Text is added and now reads: “Specifically, for the occurrence of soft tissue injuries, the subjects were asked to mark the frequency of injury in each body location (i.e., toes, foot, ankle, tibia, knee) and the type of injury (i.e., sprain, strain, rupture of the muscle/ligament/tendon, contusion abrasion, bur-sitis/tendinitis) which occurred over the last years. Dancers reported bone injurie for the whole dance-career by marking the number of fractures and/or stress fractures on each location of the body.”  (please see highlighted text in the last paragraph of the Variables subsection)

- Were anthro variables only collected once during the study? If so, depending on the period of recall for the injuries reported, authors need to comment on how current body composition related to a retrospective injury. How can the authors be sure that the dancer had a similar physique at the time of injury as what they did at the time of taking anthro measures in the current study? This is perhaps a significant limitation also affecting the correlation of this factor to injury.

Response: Thank you. Indeed, there is a certain possibility that anthropometric/body composition status at the moment of testing was not identical to dancers’ status at the moment of injury occurrence. We agree that this is certain limitation of the study, and we noted it in the Study limitations section. Text reads: “Finally, anthropometrics/body composition were measured once, while injury status was reported for previous period. Therefore, there is a certain possibility that injury occurred in a moment when dancers’ had different anthropometric/body composition characteristics than those evidenced at measurement, and this was particularly possible for bone-injuries that were reported for the whole dance-career. ” (please see subheading 4.6).

Results:

- In line 177, suggest amending to read "(68% of dancers)" and removing 'reported this type of injury'.

Response: Thank You, the text is amended.

- Define 'rupture' - ie. of ligament or tendon? Line 178

Response: Thank You, the explanation is added  (e.g. muscle/ligament/tendon)

- In line 181, suggest amending to read "(32% of dancers)" and remove 'reported injury on these two body parts'.

Response: Thank You for the comment, the excess text has been removed.

- In line 194, the results have been interpreted as dancers with longer dancing experience reported more soft tissue injury, however Table 2 shows that the injured dancers had mean experience of 10.73 years and non-injured had 11.65 years. This suggests that those with less experience were injured. Needs correction or clarification.

Response: Thank you for noticing it. The numbers in Table were not correct, it is corrected now.

Discussion:

- In line 252, avoid the use of 'first person'. Replace "we" with third person terminology.

Response: Thank You, the text is amended and now reads: “At the same time, such an explanation is not probable in our study simply because of the differences between ballet dance and the contemporary dance observed here”

- The final comments made in section 4.1, lines 252-260 require a reference.

Response: Thank you. References are added, and text now reads: “However, the improvement in technique occurs even later in the career, which conse-quently decreases the stress in the musculoskeletal system [38,39]. At the same time, dancers involved in modern dance are constantly exposed to more complex dance techniques and perform more demanding routines and more complex choreographies, which consequently increases the risk for injury occurrence irrespective of the im-provement in their dancing technique over time [40,41].”

References:

  1. Nilsson, C.; Leanderson, J.; Wykman, A.; Strender, L.-E. The injury panorama in a Swedish professional ballet company. Knee Surgery, Sports Traumatology, Arthroscopy 2001, 9, 242-246.
  2. Novosel, B.; Sekulic, D.; Peric, M.; Kondric, M.; Zaletel, P. Injury occurrence and return to dance in professional ballet: prospective analysis of specific correlates. International journal of environmental research and public health 2019, 16, 765.
  3. Ursej, E.; Sekulic, D.; Prus, D.; Gabrilo, G.; Zaletel, P. Investigating the Prevalence and Predictors of Injury Occurrence in Competitive Hip Hop Dancers: Prospective Analysis. Int J Environ Res Public Health 2019, 16, doi:10.3390/ijerph16173214.
  4. Carey, K.; Moran, A.; Rooney, B. Learning Choreography: An Investigation of Motor Imagery, Attentional Effort, and Expertise in Modern Dance. Front Psychol 2019, 10, 422, doi:10.3389/fpsyg.2019.00422.

- The final comment made in section 4.2, lines 279-281 requires a reference.

Response: Thank You, additional references are added:

  • Steinberg, N., Siev-Ner, I., Peleg, S., Dar, G., Masharawi, Y., Zeev, A., & Hershkovitz, I. (2012). Extrinsic and intrinsic risk factors associated with injuries in young dancers aged 8–16 years. Journal of sports sciences, 30(5), 485-495.
  • Kiesel, K., Plisky, P. J., & Voight, M. L. (2007). Can serious injury in professional football be predicted by a preseason functional movement screen?. North American journal of sports physical therapy: NAJSPT, 2(3), 147.

- Line 320, remove additional full stop before "Low" (.Low)

Response: Thank You for noticing, the full stop is removed.

- In line 337, suggest removing "(please see results...)"

Response: Thank You, the text is removed.

- In section 4.6, the authors should avoid inferring that causation can be determined from their cross-sectional study. Whilst the discussion of these factors is plausible, this limitation should be reworded to ensure that it is known that a causal relationship has not been established from the cross-sectional nature of the study.

Response: Thank you. We amended the text accordingly.

- An additional limitation regarding the link between injury data (depending on the period of recall) and current anthro measures needs to be acknowledged, even despite the fact that a valid instrument was used to collect injury data.

Response: The limitation is added, and text reads: “Finally, anthropometrics/body composition were measured once, while injury status was reported for previous year. Therefore, there is a certain possibility that injury occurred in a moment when dancers’ had different anthropometric/body composition characteristics than those evidenced at measurement.”

Thank you for your suggestions and comments.

Staying at your disposal

Reviewer 2 Report

Abstract:

Add study design in the abstract

The methodology should mention the measurement and protocol procedure,

The keywords should be according to MeSh

Introduction.

The study rationale is not clear,

What are the gaps in the previous studies, and how will the current study overcome those problems?

Methodology

-sample size calculation requires to explain, formula or software?

Results:

The ANOVA report should be this format (F(2,27) = 4.467, p = .021)

Discussion

Should be more critical in comparison with previous studies and report your explanation of why your results were in line with other studies.

The conclusion is too long; please summarize it

Author Response

Abstract:

Add study design in the abstract

Response: Thank You for your comment, the cross-sectional design of the study is added in the abstract.

The methodology should mention the measurement and protocol procedure,

Response: Thank You, the sentence is added and it reads: “The measurements were obtained during the pre-competition period for each specific dance discipline and were measured by experienced technician”

The keywords should be according to MeSh

Response: Thank You, the keywords are amended according to MeSh.

Introduction.

The study rationale is not clear,

Response: Thank You for your suggestion, we explained it clearly why is important to investigate this issue. The text now reads: “From the previous literature overview, it is clear that musculoskeletal injuries are important health-related problems in dance [2]. Additionally, due to the specificity of dance as an aesthetic sport, studies confirmed a high prevalence of energy deficiency and LEA in dance disciplines [18,19]. Finally, studies in other sports found evidence that LEA is a risk factor for injury occurrence in sports [26]. On the other side, there is a clear lack of research examining the association between LEA and injury occurrence among professional top level modern dancers from various disciplines. Therefore, the main aim of this study was to determine associations between musculoskeletal injuries and (low) energy availability among professional dancers. etc” (please see last paragraph of the Introduction)

What are the gaps in the previous studies, and how will the current study overcome those problems?

Response: Thank You. We explained more clearly in the study rationale what were the gaps in the previous study (please see comment above for more details).

Methodology

-sample size calculation requires to explain, formula or software?

Response: Thank you, details are added and text reads: “On the basis of number of registered modern dancers in DAS, confidence of 95%, error of 0.05, and known injury occurrence of 10%, the required sample for this research was 71 (Statistica, Tibco Software Inc., Palo Alto, CA, USA). However, due to the fact that our study was done in a period of COVID-19 pandemic (later 2021 – early 2022), and included direct measurement of anthropometrics and physical activity, please see later for details), we were not able to collect the specified number of participants.” (please see Participants subsection).

Results:

The ANOVA report should be this format (F(2,27) = 4.467, p = .021)

Response: Thank You for your comment, the results are amended accordingly.

Discussion

Should be more critical in comparison with previous studies and report your explanation of why your results were in line with other studies.

Response: In this version of the manuscript we tried to be more critical to previous findings and our results, and text is amended in some parts accordingly. For more details please see highlighted parts of the Discussion section. Thank you!

The conclusion is too long; please summarize it

Response: Thank You for your suggestion, the text is shortened significantly. Originally, the Conclusion was about 300 words long, and now is 170 words in total (please see Conclusion chapter for more details).

Once again, thank you for your suggestions and comments.

Staying at your disposal

Round 2

Reviewer 2 Report

Accepted